# Detecting Dairy Cow Behavior Using Vision Technology

John McDonagh [1,*], Georgios Tzimiropoulos [2], Kimberley R. Slinger [3], Zoë J. Huggett [3], Peter M. Down [4] and Matt J. Bell [5]

1 Jubilee Campus, School of Computer Science, University of Nottingham, Nottingham NG8 1BB, UK
2 School of Electrical Engineering and Computer Science, Queen Mary University of London, London E1 4NS, UK; g.tzimiropoulos@qmul.ac.uk
3 Sutton Bonington Campus, School of Biosciences, University of Nottingham, Sutton Bonington LE12 5RD, UK; kimberley.slinger@nottingham.ac.uk (K.R.S.); Zoe.Huggett3@nottingham.ac.uk (Z.J.H.)
4 Sutton Bonington Campus, School of Veterinary Medicine and Science, University of Nottingham, Sutton Bonington LE12 5RD, UK; peter.down@nottingham.ac.uk
5 Agriculture Department, Hartpury University, Gloucester GL19 3BE, UK; matt.bell@hartpury.ac.uk
* Correspondence: john.mcdonagh@nottingham.ac.uk

**Abstract:** The aim of this study was to investigate using existing image recognition techniques to predict the behavior of dairy cows. A total of 46 individual dairy cows were monitored continuously under 24 h video surveillance prior to calving. The video was annotated for the behaviors of standing, lying, walking, shuffling, eating, drinking and contractions for each cow from 10 h prior to calving. A total of 19,191 behavior records were obtained and a non-local neural network was trained and validated on video clips of each behavior. This study showed that the non-local network used correctly classified the seven behaviors 80% or more of the time in the validated dataset. In particular, the detection of birth contractions was correctly predicted 83% of the time, which in itself can be an early warning calving alert, as all cows start contractions several hours prior to giving birth. This approach to behavior recognition using video cameras can assist livestock management.

**Keywords:** dairy cows; computer vision; behaviors; monitoring; management

## 1. Introduction

At a time when the general public has concerns about how livestock are managed and their welfare, tools that can improve animal welfare standards and increase the public acceptance of farming are required. In recent years, the expectation has been for each stockperson to look after more animals, as input costs (including labor) have increased and finding skilled farm workers has become more challenging, and with the increased size of the average dairy herd. With these challenges have come high-quality digital camera systems that provide 24 h video surveillance capabilities, and the opportunity for farmers to monitor their livestock remotely and whilst carrying out other farm tasks. The use of cameras to monitor animals and their behaviors manually has been available for decades, with animal behavior and welfare concerns commonly directed at housed livestock production, such as dairy cows [1,2]. The monitoring of animals is essential for their welfare and survival [3].

Automated image analysis techniques have developed that allow continuous monitoring during the day and night, and require no prior training by the user other than interpreting the output. Such continuous monitoring is not possible for a stockperson. Recent technological advances in the field of computer vision based on the technique of deep learning [4,5] have emerged which now make automated monitoring of video feeds feasible. Computer vision combined with artificial intelligence (neural networks) can be used for a number of animal monitoring tasks such as recognizing the type of animals (recognition), detecting where the animals (and any other objects of interest) are located in the image (detection), localizing their body parts, and even segmenting their exact shape

(silhouette) from the image. Furthermore, adaptations of neural networks for analyzing video can be used for a number of tasks such as recognition of specific animal behaviors (e.g., standing, lying, walking, eating, and drinking) [6]. Major benefits of image analysis are that it does not rely on human interpretation or intervention, transponder attachments or invasive equipment (e.g., boluses and collars). Furthermore, it may provide more information compared to other monitoring systems at a relatively low cost. However, the technology does rely on obtaining a large number of high-quality images. The need for high-quality image datasets for agricultural solutions has been recognized by others [7]. Vision-based monitoring can not only detect and track individuals but also groups of animals (i.e., herd, flock or mother with offspring). Vision technology that can continuously monitor individual animals can potentially provide an objective assessment of an abnormal behavioral state to allow early intervention and improved awareness by a stockperson.

The objective of this study was to investigate using existing image recognition techniques to predict the behavior of dairy cows. This study collected a large number of high-quality video images for a range of cow behaviors. Such a dataset was found to be lacking but was required in the current study to train a computer vision model.

## 2. Materials and Methods

Approval for this study was obtained from the University of Nottingham animal ethics committee before commencement (approval number 151, 2017).

### 2.1. Data

Video cameras (5 Mp, 30 m IR. Hikvision HD Bullet; Hangzhou, China) were used to record Holstein–Friesian dairy cows at the Nottingham University Dairy Centre (Sutton Bonington, Leicestershire, UK) prior to calving. Cameras were recording at 20 frames per second, with a frame width of 640 pixels and height of 360 pixels. Three calving pens with two surveillance cameras looking into each pen were used to obtain 24 h video footage of 46 individual cows between April and June 2018. Both cameras on each pen allowed full coverage of the area (10 m × 7 m) and were approximately at a 45-degree angle looking into the pen at a height of 4 m. Each calving pen holds a maximum of eight cows. Several days prior to calving, each cow was moved into one of the three calving pens so that the entire calving process could be monitored.

### 2.2. Image Annotation

The video recording for each cow was annotated from 10 h before calving by three observers using custom-made scripts in the PyTorch 1.5 framework to label video clips. The PyTorch framework was used as it allows several steps in the processing of images to be carried out, such as behavioral annotations, video segmentation and model development using the Python programming language as discussed below. The start of the observation period was determined as 10 h from when the calf was fully expelled at birth using the video recording. Seven behaviors were recorded (Table 1).

**Table 1.** Studied behaviors and their description.

| Behavior | Description |
|---|---|
| Stand | The cow is still on all four legs |
| Lie | The midway transition of when the cow is about to lie down to when it starts to rise again |
| Walk | Movement of more than two steps |
| Shuffle | Cow circles on the spot or moves slightly with a step or two |
| Contractions | Visible straining while lying down |
| Eating | Cow puts its head through the feeding barrier until the moment it pulls its head back out from the feeding barrier |
| Drinking | Head is over the water trough and regular head movement towards the trough |

A total of 19,191 individual behavioral observations were obtained from all 46 cows. For the analysis, 15 video clips of each behavior that ranged between three to ten seconds were extracted from individual cow footage to provide a total of 3969 video clips for analysis (Table 2). If there were more than 15 video clips, then they would be evenly sampled from available data. There were 248–686 video clips for each behavior for training and validation. To ensure accuracy of video annotation and subsequent behavioral video clips extracted, each behavioral video clip was checked by a single trained observer to be correctly labelled and any errors corrected if required.

**Table 2.** Number of video clips for each behavior class in the training and validation datasets.

| Label | Behavior | Training | Validation | Total |
|---|---|---|---|---|
| 1 | Stand | 552 | 134 | 686 |
| 2 | Lie | 522 | 135 | 657 |
| 3 | Walk | 496 | 134 | 630 |
| 4 | Shuffle | 518 | 134 | 652 |
| 5 | Contractions | 501 | 112 | 613 |
| 6 | Eating | 392 | 91 | 483 |
| 7 | Drinking | 205 | 43 | 248 |
| | Totals | 3186 | 783 | 3969 |

The output of the behavior annotations from each video clip was described in a $N*3$ matrix, where $N$ is the total number of behaviors in the video (Table 3). Start and end frames for annotated behaviors are recorded for each video clip. Each of the retained video clips were cropped to remove excessive background and to focus on a single cow (Figure 1).

**Table 3.** Example matrix of behavior annotations.

| Start Frame | End Frame | Behavior Label |
|---|---|---|
| 553602 | 556724 | 7 |
| 556725 | 557555 | 2 |
| 557556 | 557697 | 4 |
| 557698 | 580880 | 1 |
| 580881 | 581004 | 4 |
| 581005 | 581077 | 1 |
| 581078 | 581157 | 4 |

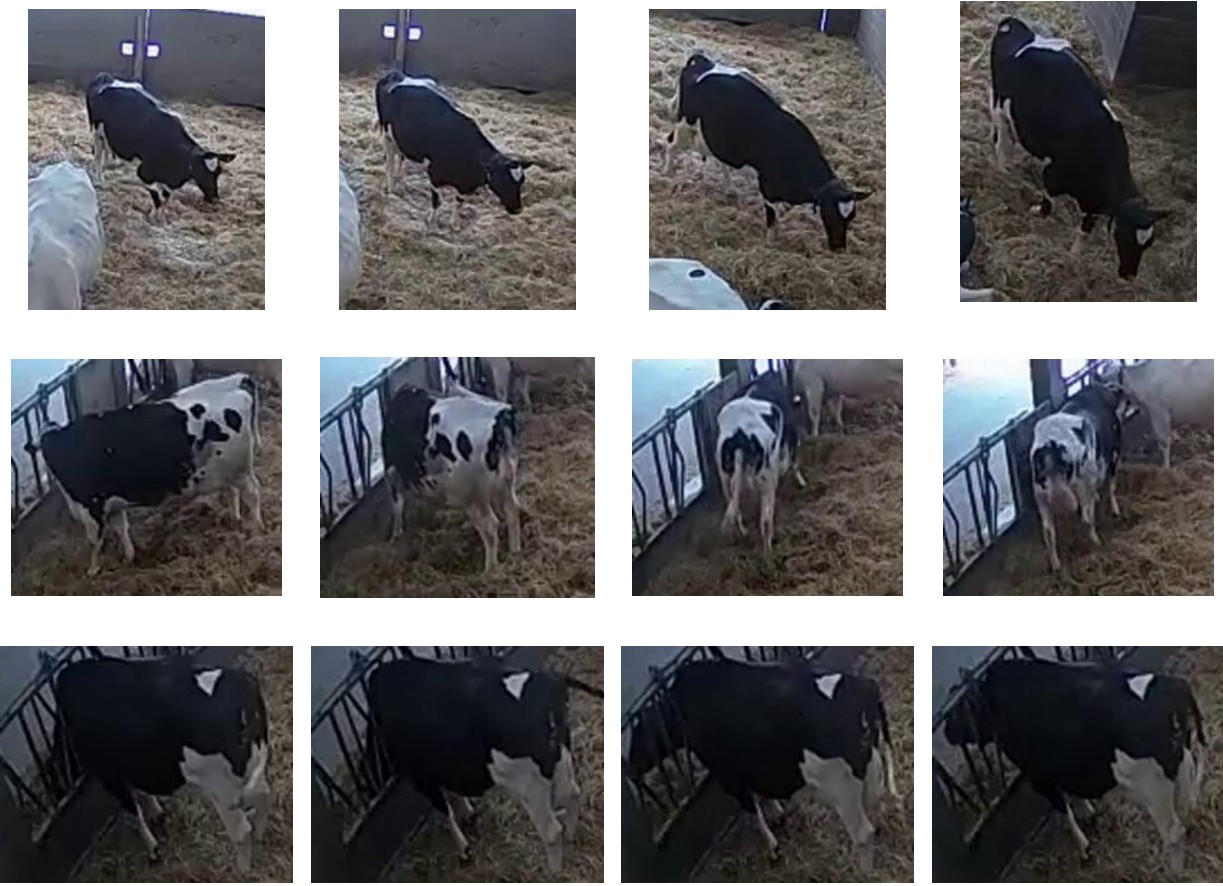

**Figure 1.** Example of cropped and scaled videos. Top row shows a cow walking, middle row shows a cow shuffling and bottom row is of a cow eating.

To be compliant with the non-local network [8], we used a fixed-size bounding box that fully covered the cow over all frames (this is to emulate [8], who used the entire frame). We used the image annotation tool ViTBAT [9] to generate the bounding boxes. The steps taken to process images for model development are illustrated in Figure 2.

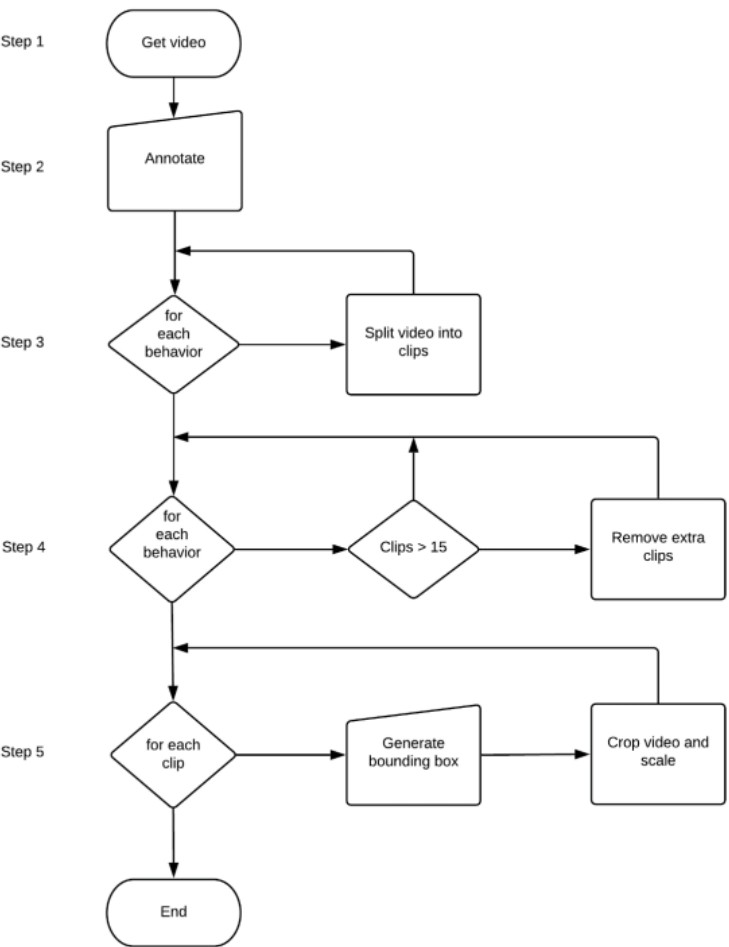

**Figure 2.** Illustration of steps in data acquisition and image processing.

### 2.3. Computer Vision Model Used for Behavior Recognition

A custom-made script in the PyTorch 1.5 framework was used to combine the behavioral matrix with cropped video images. This was performed using a non-local network [8] using the ResNet-50 architecture [10]. Further detailed explanations are discussed in prior research [8,10]. As shown in Equation (1), the non-local block computes the response at a position as a weighted sum of the features at all positions in the input feature maps and is defined as follows:

$$y_i = \frac{1}{C(x)} \sum_{\forall j} f(x_i, x_j) g(x_j), \tag{1}$$

where $x$ is the input features, $y$ is the output features (same size as $x$), $i$ is the current position of interest, $j$ enumerates over all possible positions, $C(x)$ is the normalization factor $C(x) = \sum_{\forall j} f(x_i, x_j)$, $g$ is a linear embedding $g(x_j) = W_g x_j$, where $W_g$ is learned weight matrix and $f(x_i, x_j)$ is a pairwise function that computes the correlations between the feature at location $i$ and those at all possible positions $j$.

The non-local network [8] is initialized using weights that are pre-trained on the Kinetics image dataset [11], which includes 400 behaviors for humans. This approach has been shown by [12] to improve action recognition accuracy by using a pre-trained initialization starting point for modelling. To decrease training and testing times, the current study used 8-frame input clips. The 8-frame clips were generated by randomly cropping out 64 consecutive frames from the training video and then keeping 8 frames that are evenly separated by a stride of 8 frames (Figure 3). Additionally, while training, the spatial size is fixed to 224 pixels squared, which is randomly cropped from a video or its horizontal flip, whose shorter side is randomly scaled between 256 and 320 pixels.

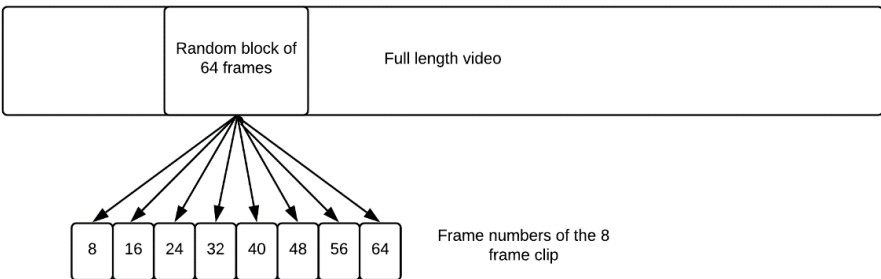

**Figure 3.** Temporal sampling of each video clip with eight evenly spaced frames being selected from a block of 64 consecutive frames.

*2.4. Computer Vision Model Validation*

To validate the performance of the model, we performed spatially fully convolutional inference as described by [8]. Briefly, the shorter side is resized to 256 pixels and 3 crops of 256x256 pixels are used to cover the entire spatial domain. The final predicted output is the average score for 10 evenly spaced 8 frame clips sampled along the temporal dimension of a full-length video (Figure 4).

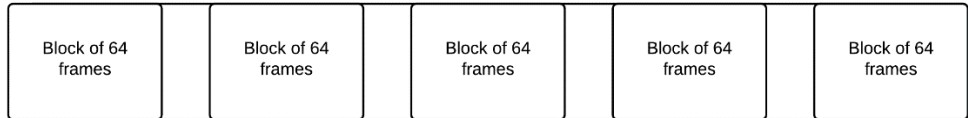

**Figure 4.** Ten clips of eight frames are sampled from blocks (64 frames) which are evenly sampled over the entire video. Each clip produces its own score, and the final output is the average of all the scores (a total of 5 blocks are shown for illustration purposes.)

## 3. Results and Discussion

Despite scientific value, pressing need and direct impact on animal health and welfare, very little attention has been paid in developing an annotated video dataset of dairy cow behaviors. Most research to date has been based on wearable accelerometer-based activity monitoring sensors [13–15]. We introduce a new large-scale video dataset for the purpose of cow behavior classification. Image banks containing a large number of high-quality (i.e., accurate and high-resolution) images for different applications are needed to develop vision-based technologies, such as behavior recognition in animals, as suggested by other studies [7]. This study showed that automated monitoring of the cow during parturition is possible, which for a high-value animal is beneficial to assist the stockperson and enhance animal welfare.

Our dataset consisted of almost 4000 video clips of individual animal behaviors, each between 3 and 10 s in length, which were on pregnant dairy cows prior to calving. There was over 9 h and 42 min of captured video data, which was split into approximately 7 h and 48 min for training and 1 h and 54 min for validation. In the field of computer vision, action recognition has been applied on humans with a high degree of success [8]. We show that the same model pre-trained on a dataset devised for human action recognition, namely Kinetics [11], can be successfully adapted to detect the behavior of dairy cows. As shown in Table 4, the accuracy of identifying contractions while lying was 83%—this in itself is sufficient enough to predict the birth of a calf, as a cow will generally start contractions approximately 1 to 2 h prior to giving birth. Standing, lying, eating and drinking behaviors all scored greater that 84% and can also help with the monitoring of animal well-being. Furthermore, changes in duration or frequency of behaviors studied may help identify abnormal behavior patterns that can assist in animal management. For example, eating and drinking can be detected with a high level of accuracy at over 90%, and these behaviors can be used to identify health problems [16].

**Table 4.** Evaluation of model predictions against validation dataset.

|  |  | Stand | Lie | Walk | Shuffle | Contractions | Eating | Drinking |
|---|---|---|---|---|---|---|---|---|
| Target [1] | No. | 134 | 135 | 134 | 134 | 112 | 91 | 43 |
| Output [2] | No. | 113 | 122 | 107 | 108 | 93 | 86 | 40 |
| Accuracy [3] | % | 84 | 90 | 80 | 80 | 83 | 95 | 93 |

[1] The target row shows how many video clips were tested for each behavior. [2] Output row shows how many behavior video clips the model classified correctly. [3] The percentage of target behavior video clips correctly classified.

As well as working with cows, the proposed computer vision approach could be adapted for other livestock species such as pigs, poultry, sheep, and horses to predict birth and identify behavior patterns or behaviors that occur over many hours, which may be missed by subjective and observational sampling. Furthermore, because the calving pen is continuously monitored, it should also be possible to detect and track the behaviors of the mother and its newborn offspring, which is not feasible using standard predictive animal monitoring applications that are currently being used by the livestock industry.

The development of behavior recognition using continuous camera surveillance within the farm environment is challenging. The current study identified several potential causes of error in computer model predictions which are limitations of current vision-based monitoring (Table 5).

**Table 5.** Potential causes of error in animal vision-based model predictions.

| Problem | Cause of Error |
|---|---|
| Pose | A cow's pose changes not only in terms of its current behavior, but also in terms of the direction it is facing from the camera. As a cow is a quadruped, this forces the model to have a much higher generalization capability when compared to bipeds such as humans. |
| Similarity | Distinguishing between two or more cows is a very difficult task even for humans. This is because cows can often have similar colors or patch patterns on their bodies. |
| Occlusion | Parts of a cow can be hidden if behind other cows, such as when all bunched up while eating. The birth of the calf can also be occluded if the cow is facing towards the surveillance camera. Cows can also be partially hidden under bedding. Cows can even have self-occlusion, where the cow's body blocks the view to other parts such as the head. Spider webs can also blur/occlude cows while the camera is in infrared night vision mode. |
| Lighting | Natural light comes through the ventilation spaces, which can produce rectangular patches over the enclosure and on the cows. Over the course of the day, the brightness of the enclosure changes. In the evening artificial lighting is used, which gives an orange tint to the enclosure. Infrared night vision is used during night-time, which turns the video footage into black and white. While the camera is in night vision mode, it focuses on the center of the pen and loses focus towards the extremities of the enclosure. Night vision also casts deep shadows off the cows that may confuse object detection. |

## 4. Conclusions

We show that computer vision can be successfully applied to predict individual dairy cow behaviors with an accuracy of 80% or more for the behaviors studied. This approach could be used for early detection of abnormal behavior in animals, birth events and the need for assistance. Computer vision technology may help a stockperson make more timely decisions based on the continuous tracking of individuals within groups of animals.

**Author Contributions:** Conceptualization, M.J.B. and G.T.; methodology, M.J.B. and G.T.; software, J.M.; validation, J.M.; formal analysis, J.M.; investigation, J.M.; resources, M.J.B.; data curation, M.J.B., K.R.S., Z.J.H. and J.M.; writing—original draft preparation, J.M. and M.J.B.; writing—review and editing, J.M., M.J.B., G.T. and P.M.D.; visualization, J.M.; supervision, M.J.B. and G.T.; project administration, M.J.B.; funding acquisition, M.J.B. and G.T. All authors have read and agreed to the published version of the manuscript.

**Funding:** This research was funded by the Douglas Bomford Trust, the Engineering and Physical Sciences Research Council and the Biotechnology and Biological Sciences Research Council.

**Institutional Review Board Statement:** The study was conducted according to the guidelines of the Declaration of Helsinki, and approved by the Animal Ethics Committee at the University of Nottingham (approval number 151, 2017).

**Informed Consent Statement:** Not applicable.

**Data Availability Statement:** The analyzed datasets are available from the corresponding author on request.

**Conflicts of Interest:** The authors declare no conflict of interest.

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
