# Peer review of "Detecting Dairy Cow Behavior Using Vision Technology"

_agriculture, doi:10.3390/agriculture11070675_

Round 1

Reviewer 1 Report

I congratulate the authors on an interesting, concise, highly relevant and well-written paper. I also congratulate the authors on making me realise I am no longer a young man who instinctively understand and follow the technological development. I apologize for ignorant questions in advance.

I am a senior scientist in the field of animal welfare, not technology. Therefore, reading the abstract, I initially wanted to decline the invitation to revise the manuscript. Then I realised that this (if I understand correctly) is exactly what I need to analyse the hours and hours of video surveillance data of dairy calves currently stored on my hard drive. My angle for reviewing this paper is hence: What information would I – not being a tech guy – need to start using vision technology? I believe this focus would increase the target audience of this paper and make it more relevant for other research areas.

As a general comment, I would say the introduction needs a paragraph (and perhaps a figure or time line) explaining in simple terms what happens from the video recording starts to the data is completely analysed, including model training, scripts, etc. I also suggest defining/explainig the terms you use, e.g. non-local and deep neural network. I looked up the references, but still did not get it.

Please also see specific comments below:

L30-32: I suggest including improved animal welfare as the main motivation for increased animal surveillance. To me this is far more important that general public concern.

L32-33: Each stockperson also has to “look after more animals” also because of increasing herds sizes, right?

L41-43: Please split this long sentence.

L50: What is meant by “high-level analysis”?

L52-55: Please split this long sentence.

L56: You state that image analysis can track groups of animals, “which is not possible using other monitoring methods”. What are “other monitoring methods”? I have tracked groups of animals using live observation and behaviour registration tools like BORIS and Ethovision. Are these not “monitoring methods”? Please elaborate/clarify.

L78-80: I assume the three observers are humans. Why three? What are their backgrounds/qualifications? Was IRR tested among the three? Did all three observe the same video material? Are “custom made scripts in PyTorch 1.5 framework” a modern way of creating an ethogram? Is “label” video clips the same as score? Please elaborate on these points.

L95: On what grounds were the 15 video clips extracted?

L96: What is “model training”?

L103-104: I do not understand “described in a matrix with start and end frames for annotated behaviors”. Please elaborate.

L104-106: Is the cropping necessary? If so, how does it affect the analyses?

2.3 and 2.4: I have no prerequisites for evaluating these sections.

2. Materials and methods: Somewhere in this section, I would like an explanation, in layman terms, how the validation between the observers and the computer vision model was conducted.

L153-154: “We introduce a new large-scale video dataset for the purpose of cow behavior classification”. Can you please elaborate on the practical application of this? Why is this dataset important to me and my work?

L189-213: Please consider rewriting this section. Although an interesting description, I do not think it fits in the discussion. I suggest (a) shortening the section, (b) converting it into a table and moving it to the introduction as a justification for the study or, ideally, (c) discussing how each of these challenges is overcome using vision technology.

Author Response

We thank the reviewer for their constructive feedback and comments. Each point has been address below and major changes made.

I congratulate the authors on an interesting, concise, highly relevant and well-written paper. I also congratulate the authors on making me realise I am no longer a young man who instinctively understand and follow the technological development. I apologize for ignorant questions in advance.

I am a senior scientist in the field of animal welfare, not technology. Therefore, reading the abstract, I initially wanted to decline the invitation to revise the manuscript. Then I realised that this (if I understand correctly) is exactly what I need to analyse the hours and hours of video surveillance data of dairy calves currently stored on my hard drive. My angle for reviewing this paper is hence: What information would I – not being a tech guy – need to start using vision technology? I believe this focus would increase the target audience of this paper and make it more relevant for other research areas.

AU: The use of old recorded video might be problematic if not of high quality. However, it has probably been annotated to a high standard and lots of images of behaviors I would presume, which is very useful. The accuracy of the computer model will be limited by the quality of the video images collected. Software is freely available online. A suggestion is contacting a computer vision researcher or authors to explore further. This is an example for calves https://www.ncbi.nlm.nih.gov/pmc/articles/PMC7795166/ but training a wearable sensor.

As a general comment, I would say the introduction needs a paragraph (and perhaps a figure or time line) explaining in simple terms what happens from the video recording starts to the data is completely analysed, including model training, scripts, etc. I also suggest defining/explainig the terms you use, e.g. non-local and deep neural network. I looked up the references, but still did not get it.

AU: A flowchart has been added as Figure 2. This shows the steps in the processing of data. Terms have been further defined in introduction and other places to help with clarity.

Please also see specific comments below:

L30-32: I suggest including improved animal welfare as the main motivation for increased animal surveillance. To me this is far more important that general public concern.

AU: This sentence has been revised.

L32-33: Each stockperson also has to “look after more animals” also because of increasing herds sizes, right?

AU: Sentence altered as suggested as “In recent years, the expectation has been for each stockperson to look after more animals, as input costs (including labor) have increased and finding skilled farm workers has become more challenging, and with the increased size of the average dairy herd.”

L41-43: Please split this long sentence.

AU: Sentence revised as suggested.

L50: What is meant by “high-level analysis”?

AU: Deleted “high-level” as not needed.

L52-55: Please split this long sentence.

AU: Sentence revised as suggested.

L56: You state that image analysis can track groups of animals, “which is not possible using other monitoring methods”. What are “other monitoring methods”? I have tracked groups of animals using live observation and behaviour registration tools like BORIS and Ethovision. Are these not “monitoring methods”? Please elaborate/clarify.

AU: This sentence has been revised as “Vision-based monitoring can not only detect and track individuals but also groups of animals (i.e. herd, flock or mother with offspring)”. Other monitoring methods was more a reference to accelerometers and GPS tracking but you can still attach a sensor to each animal.

L78-80: I assume the three observers are humans. Why three? What are their backgrounds/qualifications? Was IRR tested among the three? Did all three observe the same video material? Are “custom made scripts in PyTorch 1.5 framework” a modern way of creating an ethogram? Is “label” video clips the same as score? Please elaborate on these points.

AU: Several people annotated the video to speed up the process but one person checked all the behavioural video clips for accuracy and corrected any errors. Once video clips of cows doing certain behaviors is segmented it is much easier to visually check them. This was done to make sure they are correct. Further explanation has been added. Yes PyTorch software and programming is a free and convenient way to do multiple tasks and check video clips for accuracy rather than comparing performance of observers. Further explanation has been added as “The PyTorch framework was used as it allows several steps in the processing of images to be carried out, such as behavioral annotations, video segmentation and model development using python programming language as discussed below” in the Material and Methods.  

L95: On what grounds were the 15 video clips extracted?

AU: Additional explanation and sentence added as “If there were more than 15 video clips, then they would be evenly sampled from available data.”

L96: What is “model training”?

AU: ‘model’ removed as not needed.

L103-104: I do not understand “described in a matrix with start and end frames for annotated behaviors”. Please elaborate.

AU: Further explanation has been added to clarify as “The output of the behavior annotations from each video clip were described in a N*3 matrix, were N is the total number of behaviors in the video (Table 3). Start and end frames for annotated behaviors are recorded for each video clip. Each of the retained video clips were cropped to   re-move excessive background and to focus on a single cow (Figure 1). To be compliant with the non-local network [9], we used a fixed size bounding box that fully covered the cow over all frames (this is to emulate [9] who use the entire frame). We used the image annotation tool ViTBAT [8] to generate the bounding boxes.” The matrix is the behavioural annotation file as now shown in Table 3.

L104-106: Is the cropping necessary? If so, how does it affect the analyses?

AU: Explanation given above and in text.

2.3 and 2.4: I have no prerequisites for evaluating these sections.

  1. Materials and methods: Somewhere in this section, I would like an explanation, in layman terms, how the validation between the observers and the computer vision model was conducted.

AU: Further description given but basically all video clips once annotated we checked to make sure they were correct for model training and validation. This is very important for model accuracy and behavioural video is easy to check once segmented into the individual behaviors.

L153-154: “We introduce a new large-scale video dataset for the purpose of cow behavior classification”. Can you please elaborate on the practical application of this? Why is this dataset important to me and my work?

AU: An additional sentence has been added to explain the need for image banks to develop solutions such as behaviour recognition as “Image banks contain a large number of high-quality (i.e. accurate and high resolution) images for different applications are needed to develop vision-based technologies, such as behavior recognition in animals, as suggested by other studies [7].”

L189-213: Please consider rewriting this section. Although an interesting description, I do not think it fits in the discussion. I suggest (a) shortening the section, (b) converting it into a table and moving it to the introduction as a justification for the study or, ideally, (c) discussing how each of these challenges is overcome using vision technology.

AU: Table 5 has been created to summarise points. These are limitations of image recognition and potential causes of error in computer vision model predictions identified in the current study. We thought this was better in the Results and Discussion rather than the introduction as a finding from this work.

Reviewer 2 Report

Overall, the study is interesting and could provide useful information.  The introduction and the Material and Methods sections are well written.

Here are some specific comments on the manuscript:

Materials and Methods

L72: What was the distance between camera and animals?

L 73-74: Please, include the stocking density in pens or pen’s dimensions.

L78: Please clarify how you determine this 10-hour prior giving birth.

L79: How observers were trained?

L80: I suggest presenting the ethogram in a table.

L96: How was done the validation process?

L98: I suggest adding single trained observer.

Results & Disussion

L216: Why 80% or more? I suggest reformulating this sentence.

Author Response

We thank the reviewer for their constructive feedback and comments. Each point has been address below and major changes made.

Overall, the study is interesting and could provide useful information.  The introduction and the Material and Methods sections are well written.

Here are some specific comments on the manuscript:

Materials and Methods

L72: What was the distance between camera and animals?

AU: Further detail has been added as “Both cameras on each pen allowed full coverage of the area (10m x 7m) and were ap-proximately at a 45-degree angle looking into the pen at a height of 4m.”

L 73-74: Please, include the stocking density in pens or pen’s dimensions.

AU: The dimensions of the pen has been added as above.

L78: Please clarify how you determine this 10-hour prior giving birth.

AU: A sentence has been added as “The start of the observation period was determined as 10 hours from when the calf was fully expelled at birth using the video recording.”

L79: How observers were trained?

AU: The observations were all check when segmenting the video into behavior clips and this made it easy to see if the behaviour video clip was accurately annotated by observers. This is mentioned at lines 105 to 107 as “To ensure accuracy of video annotation and subsequent behavioral video clips extracted, each behavioral video clip was checked by a single trained observer to be correctly la-belled and any errors corrected if required.” The programming used allows multiple steps in the processing to be done as shown in figure 2, which also allows checking of video clips that are used for model training and validation. It is easy to visual inspect video clips to ensure they are correct and change if not.

L80: I suggest presenting the ethogram in a table.

AU: Table 1 added as suggested.

L96: How was done the validation process?

AU: The model validation is described in section 2.4. Further detail added.

L98: I suggest adding single trained observer.

AU: This has been changed as suggested.

Results & Disussion

L216: Why 80% or more? I suggest reformulating this sentence.

AU: This sentence has been reworded as suggested to “We show that computer vision can be successfully applied to predict individual dairy cow behaviors with an accuracy of 80% or more accuracy for the behaviors studied”. The lowest accuracies in prediction were walking and shuffle at 80%, with other behaviors being more accurately predicted.

Reviewer 3 Report

The study addresses an innovative way of detecting on-farm cows' behavior. The study is well designed but I believe it can be improved to reach scholars from knowledge fields other than computer science. For instance, the study limitations due to video recording poor quality is a reality for on-farm observations, and it should be pointed out in the Introduction session. There are plenty of references that state that issue.

I also believe the discussion should be improved. Nevertheless, the paper presents a novel way of using video footage to predict cows' behavior and help on-farm decision-making. My suggestions and questions are in the attached file. I am looking forward to seeing the corrected version.

Author Response

We thank the reviewer for their constructive feedback and comments. Each point has been address below and major changes made.

The study addresses an innovative way of detecting on-farm cows' behavior. The study is well designed but I believe it can be improved to reach scholars from knowledge fields other than computer science. For instance, the study limitations due to video recording poor quality is a reality for on-farm observations, and it should be pointed out in the Introduction session. There are plenty of references that state that issue.

AU: Sentence regarding quality of video footage has been added to the introduction as suggested.

I also believe the discussion should be improved. Nevertheless, the paper presents a novel way of using video footage to predict cows' behavior and help on-farm decision-making. My suggestions and questions are in the attached file. I am looking forward to seeing the corrected version.

AU: Comments received in an attached document have each been addressed and listed below.

Line 78 – Replace giving birth with calving

AU: Changed as suggested

Section 2.3 – Add a flowchart of the calculations to make the method clearer

AU: Flowchart added as figure 2.

Line 125 – Provide a table with the dynamic variables extracted from the video/picture sequence of the cows studied behaviors and their subsequent description

AU:  Based on comments from all reviewers further detail has been added to explain terms and data processing in the Material and Methods.

Line 129 – Depict  “The 8-frame clips were generated by randomly cropping 129 out 64 consecutive frames from the training video and then keeping 8 frames that are 130 evenly separated by a stride of 8 frames” in a picture as shown by reference [8]

AU: Added as Figures 3 and 4 to help the description of the methods used.

Section 2.4 Please present more details to the methods

AU: Further detail have been added to section 2.4 to describe the methods.

Line 154 – I believe you should address this real limitation in the introduction.

AU: This sentence has been moved to the introduction as suggested.

Line 186 – It is not clear if you are pointing out these items as limitations of the study or those conditions need improvement to use the proposed system. Please clarify.

AU: An additional sentence has been added to clarify that these are potential reasons for error in model predictions as “The current study identified several potential causes of error in computer model predictions, that are limitations of current vision-based monitoring (Table 5)”. This has been put in a table as suggested by another reviewer.